# *Cryptosporidium* life cycle small molecule probing implicates translational repression and an Apetala 2 transcription factor in macrogamont differentiation

**Muhammad M. Hasan[1,2¤], Ethan B. Mattice[1,2], José E. Teixeira[1], Rajiv S. Jumani[1,2], Erin E. Stebbins[1], Connor E. Klopfer[1], Sebastian E. Franco[1], Melissa S. Love[3], Case W. McNamara[3], Christopher D. Huston[1,2]***

**1** Department of Medicine, University of Vermont Larner College of Medicine, Burlington, Vermont, United States of America, **2** Cell, Molecular, and Biomedical Sciences Graduate Program, University of Vermont, Burlington, Vermont, United States of America, **3** Calibr at Scripps Research, San Diego, California, United States of America

¤ Current address: Division of Infectious Diseases, Departments of Medicine, and Molecular Microbiology, Washington University School of Medicine, St. Louis, Missouri, United States of America

* christopher.huston@uvm.edu

**Data Availability Statement:** RNA seq data used in this is available from the NCBI sequence read

## Abstract

The apicomplexan parasite *Cryptosporidium* is a leading cause of childhood diarrhea in developing countries. Current treatment options are inadequate and multiple preclinical compounds are being actively pursued as potential drugs for cryptosporidiosis. Unlike most apicomplexans, *Cryptosporidium* spp. sequentially replicate asexually and then sexually within a single host to complete their lifecycles. Anti-cryptosporidial compounds are generally identified or tested through in vitro phenotypic assays that only assess the asexual stages. Therefore, compounds that specifically target the sexual stages remain unexplored. In this study, we leveraged the ReFRAME drug repurposing library against a newly devised multi-readout imaging assay to identify small-molecule compounds that modulate macrogamont differentiation and maturation. RNA-seq studies confirmed selective modulation of macrogamont differentiation for 10 identified compounds (9 inhibitors and 1 accelerator). The collective transcriptomic profiles of these compounds indicates that translational repression accompanies *Cryptosporidium* sexual differentiation, which we validated experimentally. Additionally, cross comparison of the RNA-seq data with promoter sequence analysis for stage-specific genes converged on a key role for an Apetala 2 (AP2) transcription factor (cgd2_3490) in differentiation into macrogamonts. Finally, drug annotation for the ReFRAME hits indicates that an elevated supply of energy equivalence in the host cell is critical for macrogamont formation.

archive (SRA) data with accession number PRJNA1073235 (https://www.ncbi.nlm.nih.gov/sra/?term=PRJNA1073235).

**Funding:** The study was supported by National Institute for Allergy and Infectious Diseases (NIAID) R21 AI130807 to CDH (https://www.niaid.nih.gov/). The funder played no role in the study design, data collection and analysis, decision to publish, or preparation of the manuscript.

**Competing interests:** The authors have declared that no competing interests exist.

## Author summary

*Cryptosporidium* species are intracellular intestinal parasites that cause prolonged diarrhea in immunocompromised people and children. Recent in vitro studies of the *Cryptosporidium* life cycle indicate the parasite undergoes three rounds of asexual replication followed by an obligatory sexual cycle. This life cycle suggests the novel possibilities of therapy or prevention by targeting sexual differentiation and reproduction. To identify drug-like inhibitors and tool compounds, we developed a screening assay for female gametocytogenesis and screened the ReFRAME library. Our screen identified selective sexual differentiation inhibitors. We then studied the impacts of inhibitor treatment on RNA expression. Comparison of our results to publicly available female-specific gene expression data confirmed that the inhibitors selectively modulated differentiation. We then found that ribosomal protein expression falls during *Cryptosporidium* female gametocytogenesis, and we experimentally demonstrated corresponding translational repression. Further analysis of promoter regions of coordinately regulated genes implicated a conserved promoter motif and an Apetala 2 transcription factor in female gametocytogenesis. This study identifies a set of tool compounds that selectively modulate *Cryptosporidium* female gametocytogenesis, provides new insights into female differentiation, and lays the groundwork for future investigations to determine if obligatory sexual reproduction is a vulnerability that can be exploited for therapy or prevention of cryptosporidiosis.

## Introduction

*Cryptosporidium* spp. are one of the most significant etiologic agents of childhood diarrhea in developing countries [1]. There is an urgent need to develop novel therapeutics, as current treatment options are inadequate for treating malnourished children and immunodeficient patients, the two populations most vulnerable to cryptosporidiosis [2]. The parasite is a member of the eukaryotic phylum Apicomplexa that is comprised of intracellular protozoan parasites. Malaria and toxoplasmosis are two other major human diseases caused by apicomplexan parasites.

The natural life cycle of *Cryptosporidium* has been deduced from electron micrographs of animal intestines obtained at defined time points after infection with the parasite and more recently using time-lapse fluorescence microscopy to follow cycles of cell division in vitro [3–5]. *Cryptosporidium* transmits through the fecal-oral route in the form of environmentally resistant oocysts. In the mammalian gut, the oocysts excyst to release sporozoites that infect intestinal epithelial cells. Initially, *Cryptosporidium* replicates asexually during which intracellular trophozoites develop into meronts that release up to 8 merozoites that can repeat this asexual cycle. After three rounds of asexual replication, merozoites differentiate into gamonts upon infecting intestinal epithelial cells. Gamonts are of two types: uninucleated macrogamonts (i.e., female gamonts) and multinucleated microgamonts (i.e., male gamonts). Male gamonts fertilize female gamonts by infecting a macrogamont-harboring epithelial cell. Meiotic cell division in the fertilized gamont produces four sporozoites that are subsequently enclosed by an oocyst wall and released from the host cell. Oocysts either excyst within the same host to cause autoinfection or are released into the environment with feces [6]. Importantly, this model of the *Cryptosporidium* life cycle states that asexual replication is limited, which is distinct from *Plasmodium* and most other Apicomplexa and predicts that cycles of asexual and sexual replication are required to achieve sustained infection through sequential rounds of autoinfection. Given this, *Cryptosporidium* sexual replication could be targeted for

drug and vaccine development, and a better understanding of *Cryptosporidium* sexual differentiation should yield new targets for intervention.

In the laboratory setting, oocysts can be excysted by treating with sodium taurocholate and the emergent sporozoites can infect numerous cell lines [7, 8]. A widely used in vitro model of *Cryptosporidium* infection involves infecting the human colorectal carcinoma cell line HCT-8 with the human and bovine pathogen *C. parvum*. In this system, after three cycles of asexual replication, the gamonts start to emerge at 36 hours post-infection (hpi) and they are the predominant parasitic form at 72 hpi [9–11]. However, fertilization does not take place within this system, and parasite replication halts after sexual differentiation [10].

Parasitophorous vacuoles containing all stages of the *C. parvum* life cycle can be stained with *Vicia villosa* lectin (VVL) [12]. Previously, we reported a protein marker for female gamonts, DNA meiotic recombinase 1 (DMC1), that can be visualized by immunostaining [9]. We used DMC1 expression as a surogate for the appearance *C. parvum* macrogamonts and developed an assay to test the activity of drug-like small molecules on sexual differentiation. Several anticryptosporidial compounds, initially identified by screening using *Vicia villosa* lectin staining, showed a diverse range of activity against macrogamont development [9]. However, compounds specifically affecting sexual development (i.e. with no impact on asexual growth) have yet to be identified.

We now report the optimization of a DMC1-detection-based, high-content imaging assay to enable library screening for the discovery of small-molecule inhibitors that specifically target *C. parvum* macrogamont development. Based on the composition and size, the ReFRAME (Repurposing, Focused Rescue, and Accelerated Medchem) library was screened to identify high-value inhibitors. The ReFRAME library is comprised of ~12,000 compounds with high potential for drug repurposing, and since the mechanism of action of many ReFRAME library compounds is known, we reasoned that this approach would enable us to gain mechanistic insights into *C. parvum* sexual differentiation. Previous screening of the ReFRAME library against *C. parvum* asexual stages revealed new anti-cryptosporidial compounds [13]. We validated and classified screening hits with dose response curves and then chose 10 compounds that selectively modulated *Cryptosporidium* DMC1 expression for follow-up experiments. Effects of compounds on *C. parvum* macrogamont differentiation were validated using RNA-seq and comparison with publicly available RNA-seq data. This analysis provided new insights into the regulation of *Cryptosporidium* sexual differentiation. These data revealed that transcriptional repression of proteins involved in ribosome biogenesis occurs during *C. parvum* macrogamont differentiation, that an elevated supply of energy equivalence from the host cell is required for gamont formation, and that an Apetala 2-type (AP2) transcription factor (cgd2_3490) likely plays a key regulatory role in the differentiation process.

## Results

### Immunostaining for *C. parvum* DMC1 precisely identifies mature macrogamonts

In the HCT-8 cell infection model, *C. parvum* gamonts emerge between 36 and 48 hpi. DMC1 is expressed in female *Cryptosporidium* gamonts, and its expression level increases as the gamonts mature, ultimately peaking at ~72 hpi [9]. DMC1$^+$ parasites were observed with an oil immersion objective as early as 42 hpi, but they were not visible at 48 hpi using a low magnification objective that was suitable for screening. In contrast, between 40% and 60% of parasites had detectable DMC1 with the screening objective at 72 hpi (Fig 1A). Assessment of the assay quality at 72 hpi to detect DMC1$^+$ parasites grown in 384-well microtiter plates revealed a very favorable Z' score [14] which was consistently > 0.7 when compared to negative control

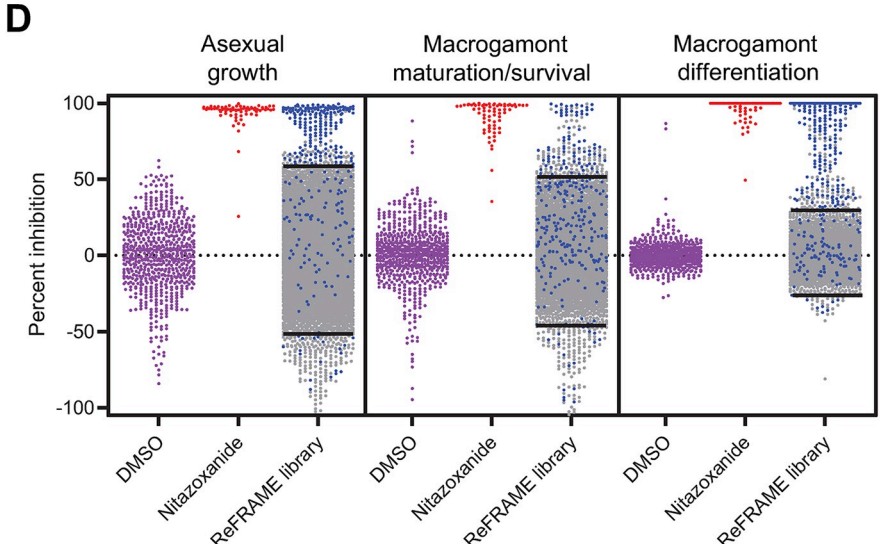

**Fig 1. Stage-specific assay and library screening. A)** Representative images of HCT-8 cells infected with *C. parvum* and stained with *V. villosa* lectin, anti-DMC1 antibody and nuclear stain. Scale bars = 100 μM. Insets show enlarged image sections demonstrating the association of DMC1 staining with parasite vacuoles. **B)** Replicate experiments showing anti-DMC1 antibody staining and automated image analysis is sensitive and specific for screening purposes. HCT-8 cell monolayers grown in 384-well culture plates were infected with *C. parvum*, and then stained for the female gamont-specific marker DMC1 at 72 hpi. The DMC1$^+$ parasite ratio was calculated for the 72 hpi timepoint, along with the z' score for each replicate. N = 56 for each replicate, 28

each for wells stained with (red) or without (blue) inclusion of anti-DMC1 antibody. **C)** Combined assay strategy showing compound addition, imaging timepoints, and the different readouts. Each readout is normalized to the readout value of vehicle treated wells from the same plate to calculate percent inhibition. **D)** Scatter dot plot showing ReFRAME library screening results for the three assay readouts. Y axis denotes percent inhibition compared to DMSO control wells. Data for negative control (DMSO) wells and positive control (nitazoxanide) wells from all plates is also shown. The cutoff values in each readout (library mean ± 2 standard deviation) for hit identification are marked by straight lines in the corresponding plot. Hits selected for further validation shown in blue fell ≥ 2 standard deviations from the library mean for at least two of three assay readouts.

wells that contained no primary antibody (Fig 1B). The reliable detection of DMC1$^+$ parasites in this miniaturized format confirmed that this was a suitable assay to screen for compounds that modulate *Cryptosporidium* macrogamont differentiation.

## Time of addition paired with multiple readouts identifies small-molecule compounds with diverse anticryptosporidial activities

For compound library screening, we used three different screen conditions to characterize compound effects on distinct aspects of the *C. parvum* life cycle: asexual proliferation, macrogamont maturation, and sexual differentiation (Fig 1C). To measure the effects on asexual growth, we added compounds at 3 hpi and enumerated the total number of parasites stained by VVL at 48 hpi and 72 hpi. The effect of compounds on macrogamont maturation and survival was assessed by adding compounds at 48 hpi and determining the number of DMC1$^+$ parasites at 72 hpi. Finally, effects on sexual differentiation were identified by adding compounds at 3 hpi and determining the proportion of DMC1$^+$ parasites at 72 hpi. Specific sexual differentiation inhibitors were predicted to decrease the number of macrogamonts without significantly altering the total number of parasites.

Using this strategy, we screened the ReFRAME compound library at a concentration of 2 μM. Screening hits for each readout were defined as compounds measuring at least two standard deviations above or below the mean for that readout of all compounds tested (Fig 1D). For follow-up, we selected compounds that were hits in at-least two different readouts. This strategy shortlisted 310 compounds (Fig 1D; blue) that detectibly modulated the *C. parvum* life cycle. Most of these hit compounds shared the same profile and either demonstrated pan-activity across all life cycle stages or specifically inhibited asexual stage proliferation; however, a subset of compounds appeared to selectively block macrogamont differentiation and maturation, which would not have been detected with the regular asexual growth assay (blue dots within two standard deviations for asexual growth assay). Requiring an effect in two different readouts omitted following up on single readout hits (grey dots in each readout above or below the two standard deviation line).

Confirmatory dose-response assays were conducted for the 310 hit compounds using each assay condition (see S1 File for all hit compounds, SMILES structures, annotation of known targets, and their final classification (discussed below); see S1 Fig for all dose-response curves). 116 (37%) of the screening hits did not reconfirm and were eliminated. Of those, 46 compounds were partial inhibitors at the highest concentrations tested. The remaining 194 screening hits (63%) were fully inhibitory in at least one assay readout at a concentration of ≤ 10 μM with the EC$_{50}$s for a given compound's most potent activity ranging from 0.027 μM to 4.1 μM.

The confirmed screen hits were classified into five categories based on the pattern of activity in the different screen conditions: 1) pan-inhibitor; 2) asexual growth inhibitor; 3) DMC1 inducer; 4) macrogamont differentiation accelerator; and 5) macrogamont differentiation inhibitor (Fig 2A–2C). Fifty-one inhibitors (26%) were active in all screen conditions with similar potency and were classified as pan-inhibitors. Nitazoxanide belonged to this category, and compounds in this category fit the general profile of ideal anticryptosporidial inhibitors for drug development. Compounds that specifically inhibited asexual proliferation were most

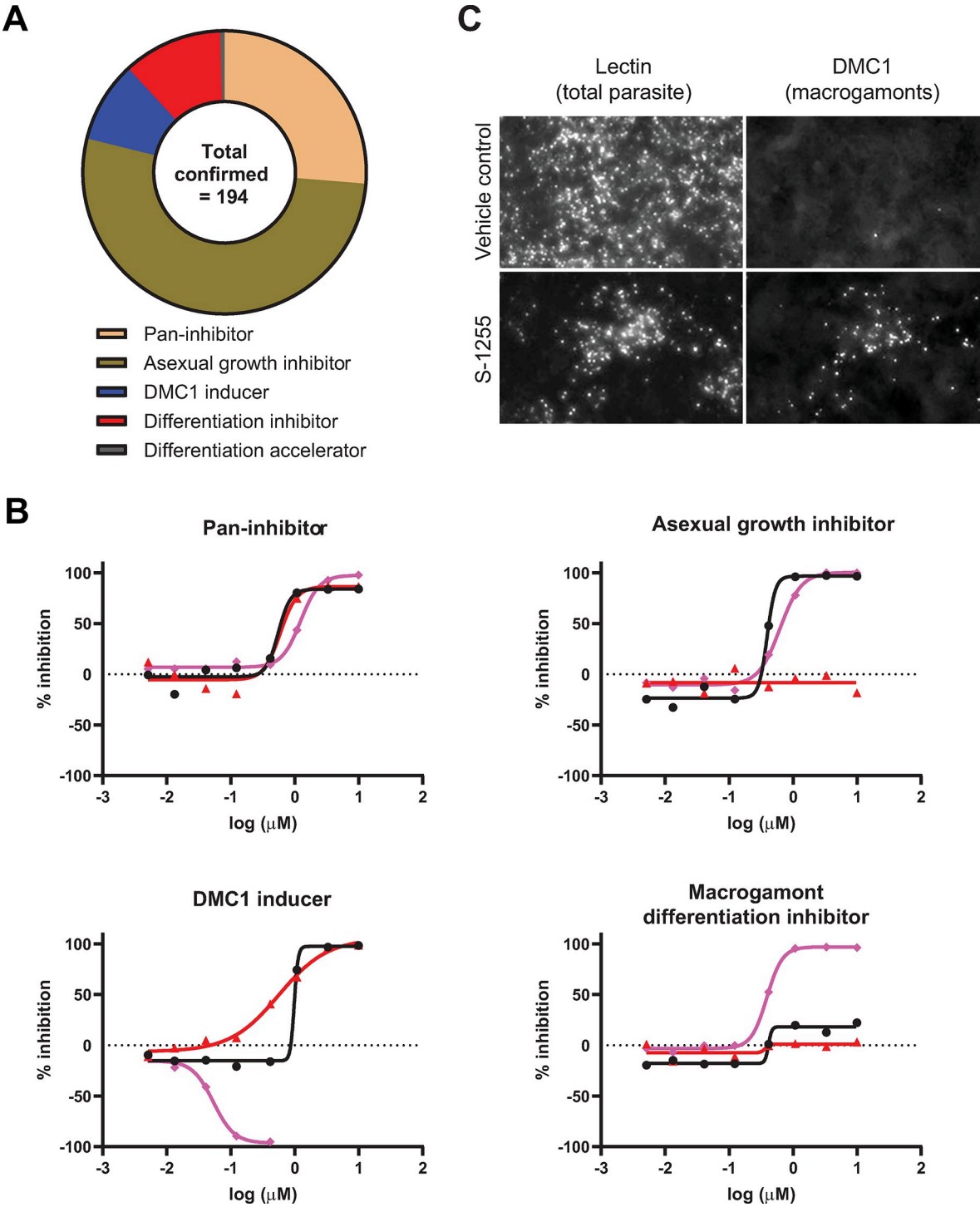

**Fig 2. Categories of inhibitors identified and example dose response curves. A)** Categorization of confirmed inhibitors. **B)** Examples of dose response curves for compounds belonging to the indicated category of inhibitor. Each point is from a single biological replicate and each curve shows one of the three readouts: Black circles and lines (effect on asexual growth); red triangles and lines (effect on existing macrogamonts); pink diamonds and lines (effect on macrogamont differentiation). **C)** Macrogamont differentiation inducer S-1255. Representative images showing *C. parvum* 48 hpi of HCT-8 cell monolayers with or without S-1255 treatment (0.55 μM). *V. villosa* lectin was used to stain all parasites, and anti-DMC1 antibody to stain macrogamonts.

**Table 1. Confirmed differentiation inhibitors and previously reported targets.** Compounds selected for follow-up experiments are in bold.

| Name | Putative target | Selectivity index$^{¥}$ | Δ maximum effect (%)$^{ß}$ |
|---|---|---|---|
| **Beloranib** | Methionine Aminopeptidase-2 (MetAP2) Inhibitor | 7.6 | 36 |
| **AGM-1470** | Methionine Aminopeptidase-2 (MetAP2) Inhibitor | 156 | 1.4 |
| **Oligomycin A** | ATP synthase inhibitor** | 2.1 | 39 |
| **Antimycin A** | Mitochondrial complex III inhibitor** | 1.1 | 20 |
| IACS-10759 | Mitochondrial Complex I Inhibitors** | 4.5 | 36 |
| PYRIDABEN | Mitochondrial Complex I Inhibitors** | 0.84 | 30 |
| **Pralatrexate** | Dihydrofolate reductase (DHFR) inhibitor* | 11.6 | 26 |
| Metoprine | Dihydrofolate reductase (DHFR) inhibitor* | 0.39 | 64 |
| **Mycophenolate** | Inosine 5'-Monophosphate Dehydrogenase (IMPDH) Inhibitors* | 4.2 | 17 |
| 6-Methylmercaptopurine | Phosphoribosylpyrophosphatate amidotransferase (PPAT) inhibitor* | 2.6 | 24 |
| Pelitrexol | Glycinamide Ribonucleotide Formyltransferase (GARTFase) Inhibitors* | 0.4 | 73 |
| **TVB-2640** | Fatty acid synthase inhibitor | ? | 57 |
| **BAY 61–3606** | Tyrosine kinase inhibitor | 11 | 27 |
| **Mubritinib** | ErbB-2 tyrosine kinase inhibitor | 4.3 | 20 |
| RWJ-46458 | Angiotensin AT1 Antagonists | 5.7 | 46 |
| OCT-1547 | RANK ligand inhibitor | 3 | 32 |
| BX-471 | CC chemokine receptor 1 antagonist | 2.5 | 32 |
| CAI orotate | PI3 kinase inhibitor/ VEGF receptor antagonist/ Calcium channel antagonist | 2.8 | 34 |
| SCHEMBL10678365 | Antidepressant | ? | 30 |
| ZD 2138 | Lipoxygenase 5 Inhibitor | 2.2 | 9 |
| FLUFYLLINE | 5 Hydroxytryptamine 2 receptor antagonist | 1.1 | 31 |

* denotes targets in nucleotide biosynthesis pathways

** denotes mitochondrial oxidative phosphorylation inhibitors

$^{¥}$ Selectivity index = $EC_{50\text{ asexual growth}}$ / $EC_{50\text{ sexual differentiation}}$

$^{ß}$Δ max effect (%) = maximum % inhibition$_{\text{sexual differentiation}}$−maximum % inhibition$_{\text{asexual growth}}$

frequently identified and comprised 102 hit compounds (53%). Classified as asexual growth inhibitors, these compounds only demonstrated detectable activity when added 3 hpi and were inactive against existing macrogamonts when added at 48 hpi. Interestingly, eighteen compounds (9%) appeared to induce macrogamont formation, since they increased the proportion of DMC1$^+$ parasites present at 72 hpi with little effect on the total number of parasites present up to a certain dose. These compounds were classified as DMC1 inducers. One compound (0.5%), the Endothelin A receptor agonist S-1255, induced DMC1 expression at 48 hpi (Figs 2C and S2). This compound was classified as a macrogamont differentiation accelerator and was unique amongst the compounds tested. Finally, based on selectivity for inhibition of macrogamont development when added at 3 hpi vs. effect on the total number of parasites (Selectivity index = $EC_{50\text{ asexual growth}}$ / $EC_{50\text{ sexual differentiation}}$ ≥ 2 and/or >20% difference in maximal effect on growth vs. differentiation), twenty-two compounds (11%) were classified as selective inhibitors of sexual development. We categorized these compounds as macrogamont differentiation inhibitors, and they were the predominant focus of our follow-up experiments (Table 1).

## Life cycle stage-specific genes are dysregulated by differentiation modulators

The compound hits from the pan-inhibitor and asexual growth inhibitor categories overlap with those previously discovered in a separate ReFRAME screen [13]. In contrast, the

modulators of sexual differentiation are novel. Based on available resources for follow-up studies, we downselected to ten differentiation modulators as chemical tools to further study the mechanisms driving macrogamont differentiation. The single differentiation accelerator (S-1255) was selected based on novelty, and nine differentiation inhibitors were selected based on a combination of commercial availability for resupply and relative selectivity for macrogamont differentiation vs. inhibition of *Cryptosporidium* growth (see Table 1). These ten compounds were used for RNA-seq studies with two goals: further validating the effects of the compounds on sexual differentiation through comparisons with publicly available RNA-seq data; and gleaning new insights into the *Cryptosporidium* differentiation mechanism. Since most of the compounds had mixed effects and inhibited parasite growth at concentrations exceeding those needed to inhibit differentiation, we first identified the concentration for RNA-seq that was most selective for macrogamont differentiation compared to growth (i.e., the concentration with the biggest difference in percent inhibition of macrogamont differentiation vs. *Cryptosporidium* growth (Figs S2 and 3A)) and the latest time point that the compounds could be added (S3 Fig).

A suitable time point for RNA-seq to discern compound effects on stage-specific gene expression was determined by conducting RNA-seq in the absence of differentiation inhibitors at 18, 36, 48, and 72 hpi (Fig 3B). Overall, very few parasite genes were differentially regulated between 18 and 36 hpi, while many genes were differentially regulated between 36 and 48 hpi, and 48 and 72 hpi. We selected the 48 hpi time point to assess dysregulation of stage-specific genes by compound exposure. Either compound or DMSO (untreated control) was added at 3 hpi, and RNA-seq libraries were prepared using samples taken at the 48 hpi timepoint (Fig 3C; 2 independent biological replicates each, except for pralatrexate (n = 1) for which library preparation failed for one replicate) (S1 Table).

As expected, compound treatment affected both host and parasite gene expression (Fig 3C). S2 File contains the transcripts per million (TPM) values of all the *C. parvum* genes identified from each sample, along with the fold change on a $\log_2$ scale (false discovery rate (FDR) $\leq$ 0.1) compared to the 48 hpi untreated control.

The expression level of *dmc1* in our initial time course RNA-seq dataset correlated well with published expression levels at different time points [9]. Except for the fatty acid synthase inhibitor TVB-2640, treatment with the differentiation inhibitors significantly reduced expression of *dmc1* at 48 hpi, as expected. Furthermore, treatment with the differentiation accelerator S-1255 stimulated *dmc1* expression at 48 hpi (Fig 3D). Collectively, these data validated the differentiation modulators as powerful chemical probes to study *Cryptosporidium* sexual differentiation.

To test whether the genes dysregulated by compound treatment were stage-specific, we compared our dataset to a previously published RNA-seq dataset of gene expression of FACS-sorted asexual and female macrogamont-stage *C. parvum* [10]. Using this dataset, we divided *C. parvum* genes into three groups: female-specific (upregulated in females compared to asexual); asexual-specific (downregulated in females compared to asexual); and unresolved (not differentially regulated between the two sorted populations) (S3 File). Following treatment with the differentiation inhibitors, asexual-specific genes were predominantly upregulated, and female-specific genes were predominantly downregulated. The reverse pattern was observed for treatment with the differentiation accelerator S-1255 (Fig 3E). No such dichotomy was observed for the unresolved gene set. We therefore concluded that the differentiation inhibitor and accelerator treatments selectively affected the transition in gene expression that occurs upon differentiation from asexual-stage parasites to female gamonts.

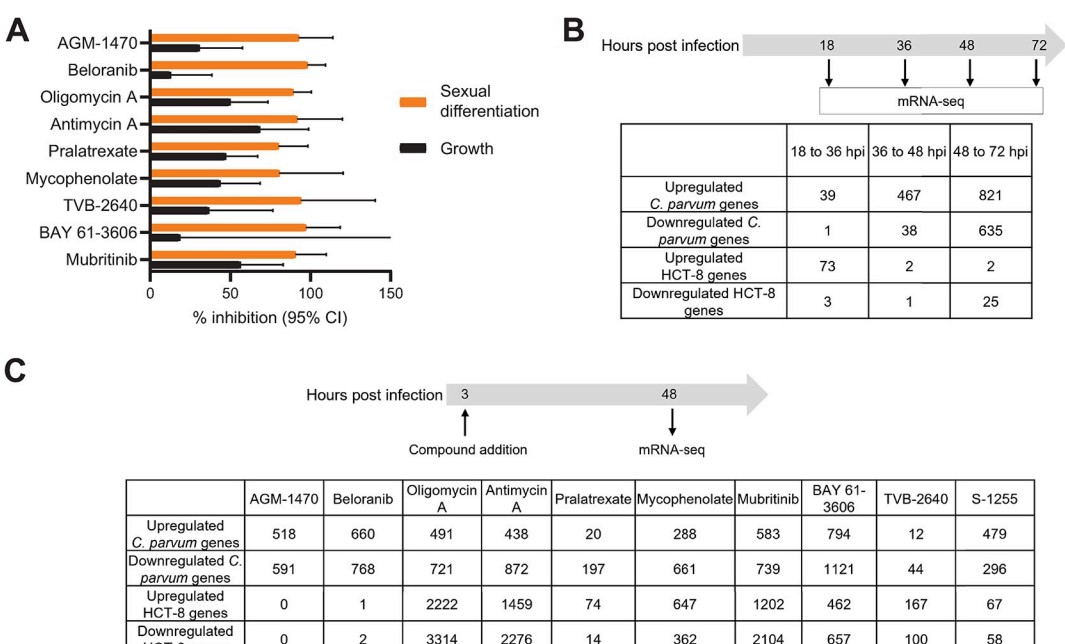

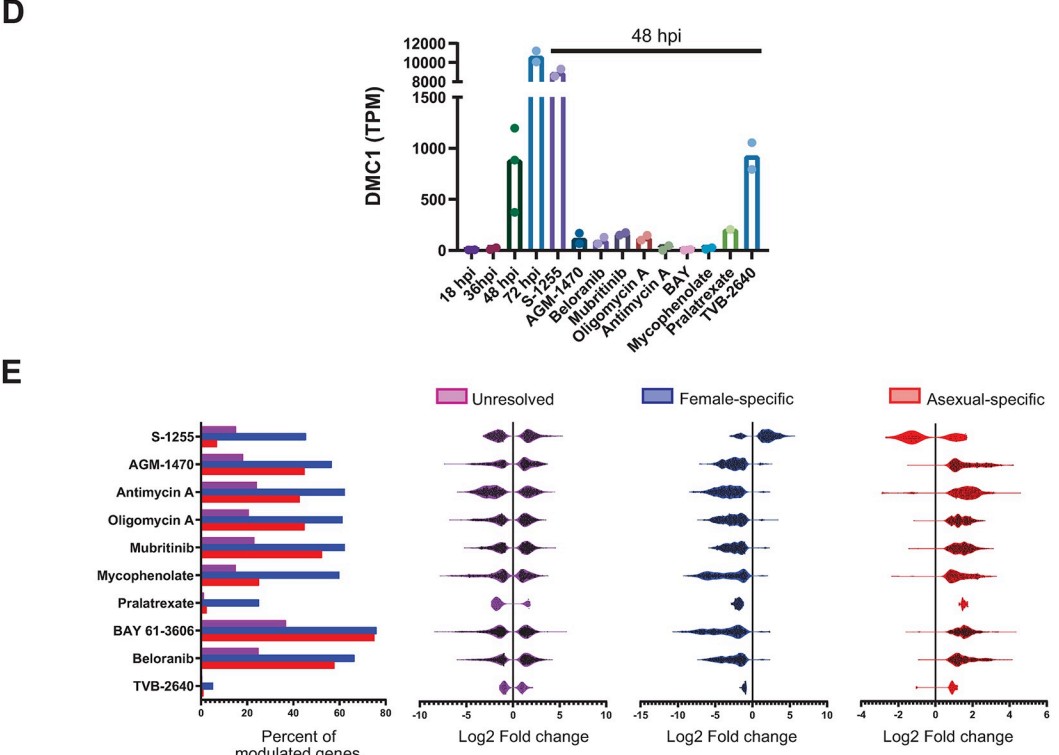

**Fig 3. Treatment with differentiation inhibitors alteres expression of a set of core genes involved in parasite differentiation. A)** Relative effects on macrogamont differentiation and growth for the indicated inhibitors using the dose found to provide greatest selectivity for differentiation (see also S2 Fig). **B)** Experimental setup for the pilot RNA-seq experiment. The number of genes dysregulated (fold change ≥2 at a False Discovery Rate of ≤ 0.1) between subsequent timepoints of the experiment are listed. **C)** Experimental setup and the number of dysregulated genes (fold change ≥2 at a False Discovery Rate of ≤ 0.1) with differentiation modulator treatments. **D)** Transcript per million (TPM) values of *dmc1* across all the RNA-seq samples. **E)** All the *C. parvum* genes were divided into three classes based on previously published RNA-seq comparison between female and asexual stages of *C. parvum* [10]. Genes upregulated in females compared to asexual stages are denoted female specific genes (668 in total) and genes downregulated in females compared to asexual stages

are denoted as asexual stage-specific genes (450 in total). All the other genes (2902 genes) are denoted as unresolved. The bar graph shows what percentage of genes from the above-mentioned classes are dysregulated (FDR ≤ 0.1 and and log2-fold change cutoff of 2) with the marked treatment at 48 hpi. Note that a higher ratio of the stage-specific genes are dysregulated compared to the unresolved class. The violin plots show the log2-fold changes in the expression of genes that meet the above-mentioned cutoff of the significantly differentially expressed genes, grouped by the mentioned gene categorization.

## Transcriptional profile of ribosomal proteins is altered with differentiation modulator treatments

We performed functional annotation enrichment analysis of the RNA-seq upregulated and downregulated gene sets for every treatment using the NIH DAVID Bioinformatics Resource (version 6.8) [15]. The upregulated genes for seven of the differentiation inhibitor treatments were highly enriched for genes encoding proteins involved in ribosome biogenesis and ribosome structure (S4 File and Fig 4A). These two pathways were also significantly enriched in the asexual-specific gene set derived from the previously published dataset (Fig 4A) [10]. Looking specifically at ribosomal proteins (RPs), we found that mRNAs encoding this class of proteins reached their peak concentration at 36 hpi and then decreased rapidly at 48 and 72 hpi. As predicted, hierarchical clustering of the expression pattern of RP-encoding genes clustered the differentiation inhibitor-treated samples (prepared from 48 hpi) with 18 hpi and 36 hpi control samples, whereas the differentiation inducer-treated samples clustered with the 48 hpi and 72 hpi control samples (Fig 4B). Taken together, these data show that repressed expression of RPs is a transcriptional signature of female gamonts of *C. parvum* and that the ten differentiation modulators tested here directly or indirectly alter this regulation.

## Transcriptional repression of ribosomal proteins is associated with global translational repression in *Cryptosporidium*

Repression of RPs likely leads to an overall reduction of protein translation. Therefore, we predicted higher protein translation by *C. parvum* during asexual growth than in mature sexual phases of the life cycle. To test this prediction, we used an alkyne analog of puromycin, O-propargyl-puromycin (OP-puro), that can be labeled using copper(I)-catalyzed azide-alkyne cycloaddition (i.e. CLICK chemistry) to image protein synthesis in *C. parvum* at various culture time points [16]. Out of focus light representing host-cell protein synthesis is a confounder but can be largely excluded because of the location of *Cryptosporidium* vacuoles which reside on the apical surface of infected cells. Parasite protein synthesis was readily detected during asexual growth at 36 hpi and blocked by cycloheximide, which interferes with eukaryotic translational elongation (Fig 4C). On the other hand, parasite protein translation at 72 hpi was minimal. Consistent with the RNA-seq data demonstrating repressed expression of ribosomal proteins in sexually mature *C. parvum*, protein synthesis peaked at 24 to 48 hpi and was low at 60 and 72 hpi when gametocytes predominate in the HCT-8 culture system (Fig 4D).

To check whether or not the transcriptional repression of ribosomal protein genes during sexual differentiation is unique in *C. parvum* among apicomplexans, we analyzed several RNA-seq datasets available in the VEuPathDB database [17] by comparing the transcriptomes of different stages of *Plasmodium*, *Toxoplasma*, and *Eimeria* [18–22]. We first identified differentially regulated genes between life cycle stages using the database's built-in differentially expressed gene selection tool and then subjected the identified genes to gene ontology enrichment analysis (S5 File). We found that ribosomal proteins are significantly enriched amongst genes with altered regulation during differentiation of *Eimeria*, *Plasmodium*, and *Toxoplasma* (Fig 4E), suggesting that this is a conserved feature of stage differentiation in apicomplexan parasites.

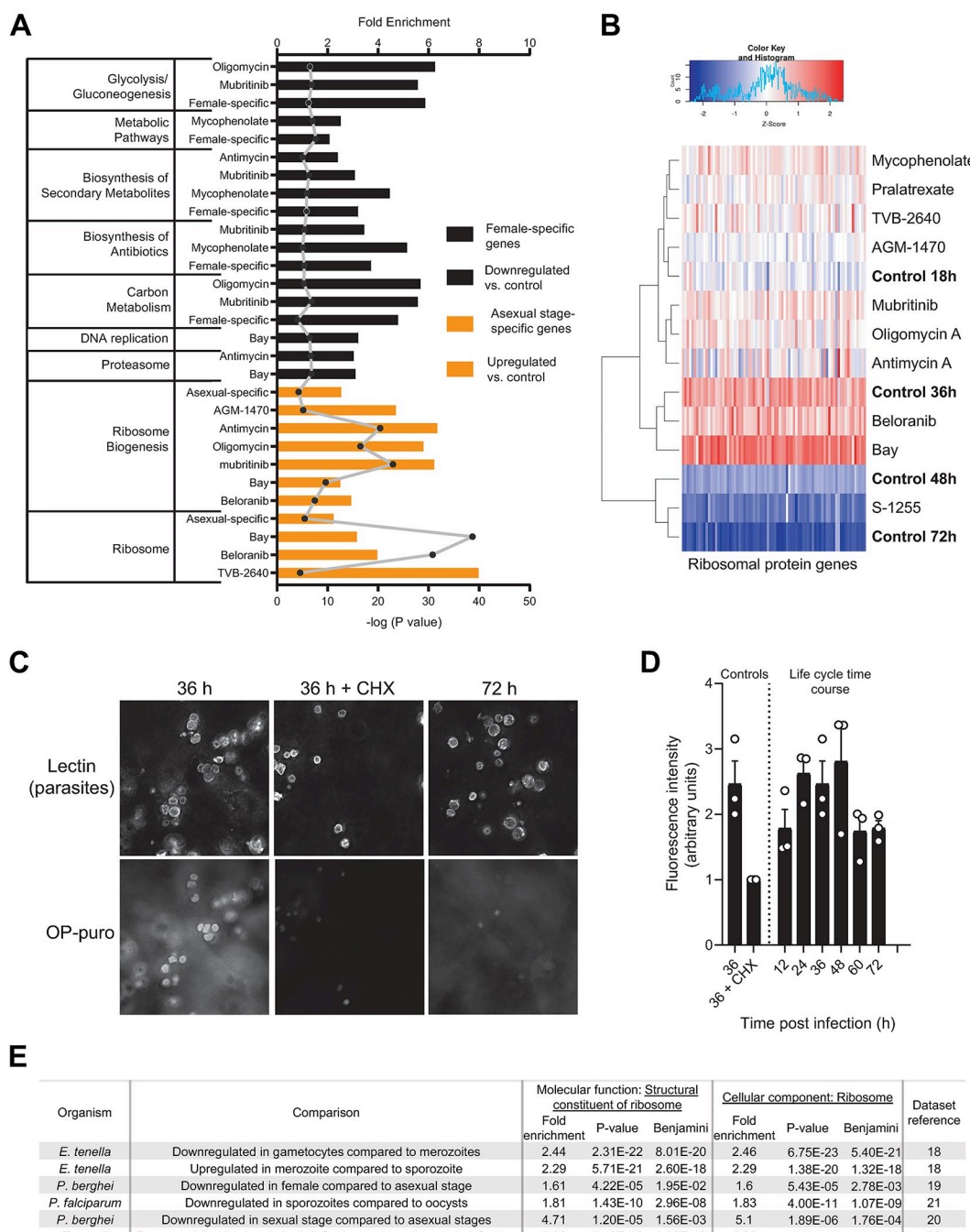

**Fig 4.** *parvum* **sexual differentiation is associated with suppression of ribosome formation and reduced protein translation. A similar transcriptional signal is found in other Apicomplexa.** *C.* **A)** A summary of DAVID functional annotation results for significantly dysregulated *C. parvum* genes with different treatments, as well as female and asexual stage-specific gene set lists taken from the previously published dataset [10]. Enrichments with an FDR of ≤ 0.01 from KEGG pathway are included only. The bars denote fold enrichment of the labeled pathway in the labeled gene set whereas points denotes the enrichment P-value. **B)** Heatmap of the average transcript per million (TPM) values of ribosomal protein genes of *C. parvum* at various timepoints (control) and in treatments with differentiation inhibitors (48 hpi). Each column represents a single gene encoding a ribosomal protein. TPM values are Z normalized across each column. Hierarchical clustering between the samples using the expression values of the ribosomal proteins was performed using the "complete" linkage algorithm and Euclidean distance metrics and is shown on the left of the heatmap. **C)** Nascent proteins in cultured *C. parvum* imaged using OP-puro. HCT-8 cells were infected with *C. parvum* and cultures were pulsed with the alkyne puromycin analog OP-puro 1 h prior to the indicated time points. Cells were then fixed, stained by CLICK-chemistry with Alexa 568-azide, and imaged by fluorescence microscopy. Treatment with the translation inhibitor cyclohexamide (CHX;

50 μM) served as a control for staining specificity. **D)** Quantified parasite protein translation vs. time of infection. The mean fluorescence intensity in the OP-puro channel of parasite vacuoles identified by *V. villosa* staining was measured. Note that out of focus light from ongoing host cell protein synthesis confounded the analysis. Data shown are the mean and SE for three independent experiments of the mean fluorescence intensity relative to the 36 h CHX control. **E)** Significant enrichment (p-value cutoff $\leq 0.5$) of the two specific gene ontology (GO) terms during gene ontology enrichment analysis of apicomplexan genes that are differentially regulated between the listed life cycle stages. Dysregulated genes are defined as genes with a log2-fold difference of $\geq 1$ at 10% false discovery rate (FDR).

## Identities and effects of differentiation modulators predict critical regulators of *C. parvum* macrogamont differentiation

**Macrogamont differentiation takes place between 36 and 48 hpi:** We assessed whether there was a temporal window when the differentiation inhibitors needed to be present to exert their maximal effect. The inhibitors were added at 3, 12, 24, 36 or 48 hpi, and the percentage of DMC1$^+$ parasites was measured at 72 hpi (S3 Fig). Six compounds were effective when added as late as 36 hpi. On the other hand, three compounds (i.e., pralatrexate, mycophenolate, and TVB-2640) lost activity if added at or after 24 hpi, suggesting that prolonged exposure was required and/or that they exclusively affect early stages of differentiation, such as commitment. Importantly, if added after 48 hpi, none of the inhibitors affected *dmc1* expression, reinforcing the interpretation that they affected sexual differentiation, and this process was distinct from gamont maturation. Nitazoxanide, a pan-inhibitor, had the same effect regardless of the timing of addition. This was consistent with the likelihood that pan-inhibitors have a general toxic effect on *C. parvum* that is independent of the stage of the parasite's life cycle.

**A heptameric promoter element controls the expression of ribosomal proteins:** Ribosomal protein genes are scattered across the *C. parvum* genome, yet their expression levels were highly correlated with each other across time points and compound treatments (Figs 4B and S4). This prompted us to search for conserved sequence signatures within the upstream and downstream untranslated regions of these genes. Differential enrichment analysis of the 5'UTR regions (1 kb maximum with removal of encroaching coding sequences) of the ribosomal protein genes identified a DNA motif, (A/T/C)GAGAC(A/G), that was highly enriched in these genes compared to all the annotated protein coding genes of *C. parvum* (Fig 5A). A very similar motif was previously reported to be enriched in RP promoters in *C. parvum* [23]. We hypothesized that this is a transcription regulatory motif and if that is true, the majority of the other genes having this motif in their promoter region might have similar expression profiles to RP genes. To test this hypothesis using our transcriptomic dataset, we determined the degree of similarity in the expression profile of all the *C. parvum* genes to the RP genes across all our samples. Indeed, in the promoters of the 100 most similarly expressed genes, this heptameric motif was found in 47 promoters (63 times total), whereas in the promoters of the 100 most dissimilarly expressed genes, it was present in only nine instances (Fig 5B; p <0.0001).

**The Apetala 2 Transcription Factor cgd2_3490 is a key regulator of *C. parvum* macrogamont differentiation:** Genes that are downregulated with differentiation inhibitors but upregulated with the differentiation inducer treatment can be considered as sexual stage-specific genes that are coregulated. Sixty-five genes were downregulated with eight of the nine differentiation inhibitor treatments but upregulated with S-1255 treatment in our differential expression analysis (Fig 6A and S6 File). We excluded TVB-2640 treated samples to select coregulated macrogamont stage-specific genes because the mRNA level of *dmc1*, a validated macrogamont stage-specific protein and the basis of our stage-specific assay, was not significantly downregulated with TVB-2640 treatment. Specific regulatory DNA sequences might control the macrogamont stage-specific expression pattern of these 65 genes. Therefore, we screened the promoter regions (maximum 1 kb upstream with removal of encroaching coding

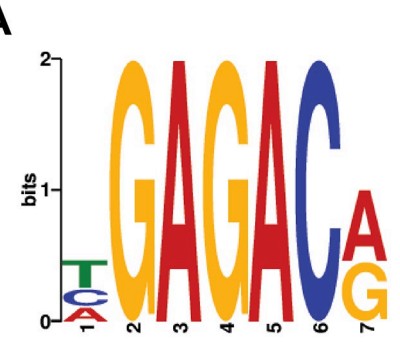

**A)** Present in
66 of 78 Ribosomal proteins
520 of 3956 *C. parvum* genes
Enrichment P-value: 1.6e-44

**B)**

| Contingency Table | "HGAGACR" positive | "HGAGACR" negative |
|---|---|---|
| Gene expression pattern similar to ribosomal proteins | 47 | 53 |
| Gene expression pattern opposoite to ribosomal proteins | 9 | 91 |

| Test | Chi-square |
|---|---|
| Chi-square, df | 35.81, 1 |
| z | 5.984 |
| P value | <0.0001 |
| One- or two-sided | Two-sided |

**Fig 5. DNA motif significantly enriched in ribosomal protein gene promoters. A)** The most significantly differentially enriched DNA motif in the promoter region of the ribosomal protein genes compared to all *C. parvum* genes. **B)** Chi-square test of the association between the presence of the DNA motif from (**A**) in the promoter region of 100 most similarly and dissimilarly expressed genes to the RP genes.

sequences) of these 65 genes for enrichment of specific DNA motifs compared to all the other *C. parvum* promoters and identified that an 8 bp motif, TGCATG(T/C)(G/A), was highly enriched in the promoters of this gene set (Fig 6B). Interestingly, an almost identical DNA motif was previously identified as the binding motif of two *C. parvum* Apetala 2 transcription factors, cgd2_3490 and cgd1_3520 [24]. Among them, the expression profile of cgd2_3490 itself mirrored that of the 65 selected genes, i.e., down regulated following differentiation inhibitor treatment and upregulated following inducer treatment (Fig 6C). Therefore, cgd2_3490 is potentially expressed during the sexual differentiation of *C. parvum* to transcriptionally activate this set of macrogamont stage-specific genes.

## Discussion

In this study, we screened the ReFRAME library, a high-value collection of clinical-stage and approved small molecules, to identify modulators that differentially affect the asexual and macrogamont stages of *C. parvum* growth and development. We reasoned that in addition to identifying compounds with potential utility as starting points for drug development, we would also identify powerful tool compounds with which to study *Cryptosporidium* sexual differentiation. Most of the compounds identified (collectively 78% of the verified hits) either affect all stages of *C. parvum* growth and development, consistent with a general toxic effect on the parasite, or preferentially affect the rapid phase of asexual growth when processes such as protein synthesis and DNA replication are expected to predominate. However, the remaining verified hits selectively affected the development of macrogamonts, suggesting that they affect one or more process that is required for sexual differentiation. Nine macrogamont differentiation inhibitors and one differentiation accelerator were selected for further characterization based on their availability and selectivity for macrogamont development vs. other stages of the parasite life cycle. We used these compounds in combination with RNA-seq to gain insights into control of *Cryptosporidium* macrogamont differentiation. We validated the RNA-seq dataset and specific impact of the compounds on macrogamont development by comparison with a publicly available RNA-seq dataset generated using flow cytometry sorted asexual

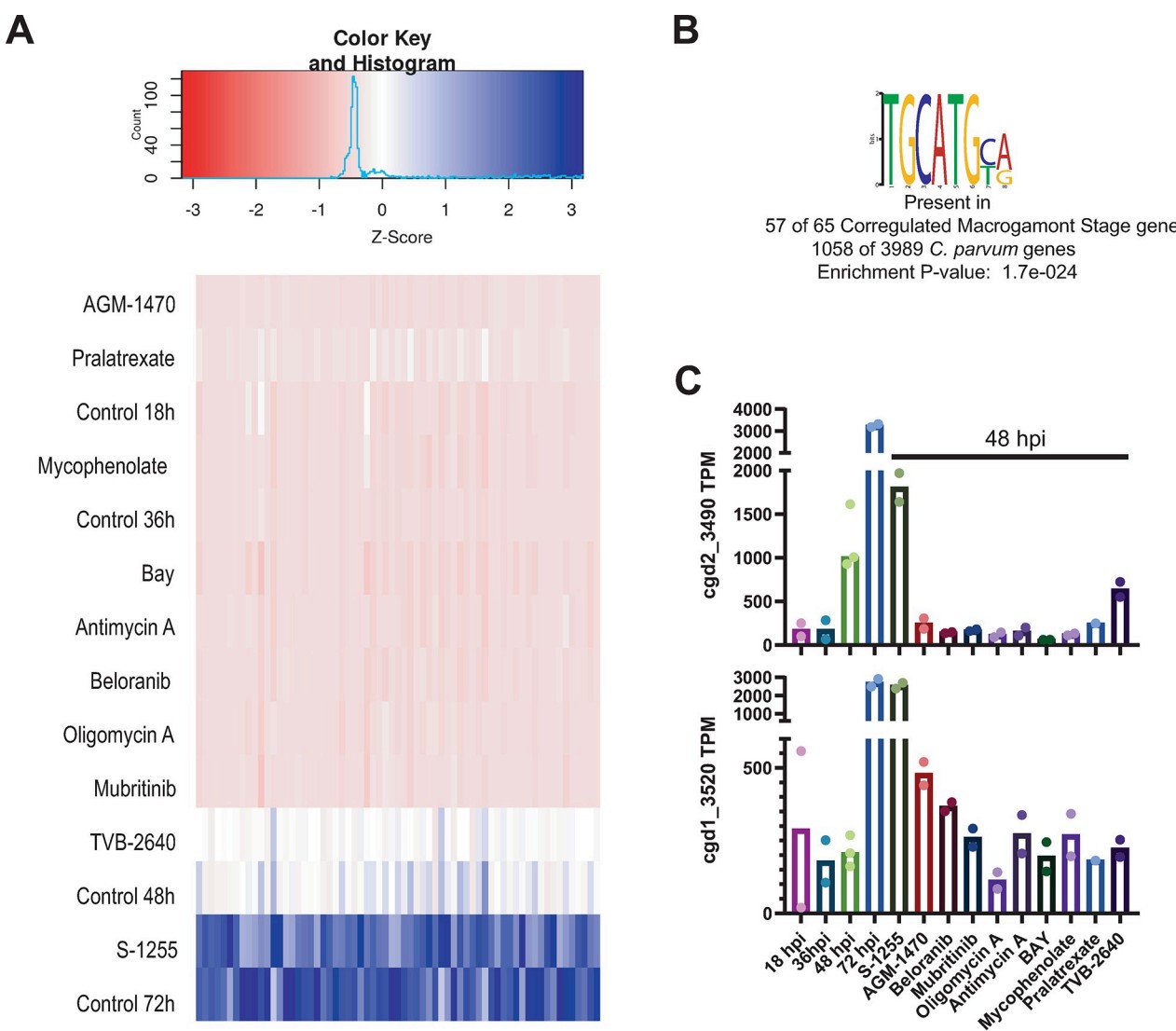

**Fig 6. Promoters of downregulated genes of *C. parvum* harbor a recognition site for two Apetala 2 transcription factors that are also dysregulated by differentiation inhibitor treatments. A)** Expression heatmap from the average TPM values from all the samples of the genes that are downregulated in eight of the nine the differentiation inhibitor treatments (except TVB-2640, which was not included in the filter process; i.e, the gene can be up, down or not differentially regulated in this treatment) and upregulated with the treatment of the differentiation inducer S-1255. Each column represents a gene from 65 genes that meets our filtering criterion. TPM values are Z normalized across each column. and upregulated with the treatment of the differentiation inducer S-1255. **B)** The most significantly differentially enriched DNA motif in the promoter region of the selected 65 genes. **C)** Expression values of the two Apetala 2 transcription factors that bind a highly similar motif like the one shown in B.

parasites and macrogamonts [10]. This analysis in combination with the putative targets of the macrogamont differentiation inhibitors highlights differences in the metabolic needs of the parasite at different stages of development, demonstrates a general program of translational repression in *Cryptosporidium* sexual differentiation that is shared by other Apicomplexa, and identified critical gene regulatory motifs and an Apetala 2 transcription factor that likely function in differentiation.

The strategy to screen the ReFRAME library across a panel of assay conditions facilitated an unambiguous binning of hit compounds into five distinct classifications. Furthermore, the $EC_{50}$ activities of the hit compounds were valuable to the classification process because a majority of the hits were active against multiple stages, albeit with different potency. Therefore,

for each compound, we considered the readout where the potency was highest as its primary mode of action.

The identities of multiple differentiation inhibitors suggests that macrogamont differentiation requires increased uptake of energy from the host cell by the parasite. Around 86% of ReFRAME compounds were developed to treat human conditions not caused by infectious agents [13]. Therefore, many ReFRAME compounds target human proteins or RNAs. As an intracellular parasite, *C. parvum* must be sensitive to altered conditions within host cells. It is therefore highly likely that validated hits from our screen may be acting indirectly on *C. parvum* by altering the host cell environment without directly affecting a parasite drug target. To that point, four differentiation inhibitors block mitochondrial oxidative phosphorylation (Table 1), but *C. parvum* does not contain a fully functional mitochondrion itself [25]. Two of these four compounds were included in our RNA-seq follow-up, and they dysregulated a high number of host genes in addition to parasite genes (Fig 3C). Interestingly, screen hits that were inhibitors of oxidative phosphorylation were exclusively classified as differentiation inhibitors. The primary function of oxidative phosphorylation is to generate energy. Could it be the case that *C. parvum* gamonts need more energy from the host cell compared to when they are growing asexually? This line of reasoning is supported further by some of our data and recently published literature. For example, it has been proposed that host-derived purine nucleotides are directly imported as ATP which fulfills the energy demand of the parasites in addition to serving as nucleic acid building blocks [26]. Amongst all screen hits, we found five compounds that inhibit host-cell purine nucleotide biosynthesis, and all of them were differentiation inhibitors. The confirmed asexual growth inhibitors also included compounds that inhibit nucleotide biosynthesis, but only pyrimidines (Fig 7). Moreover, in a separate host-cell platform that supports complete gamont maturation and fertilization of *C. parvum*, oxidative

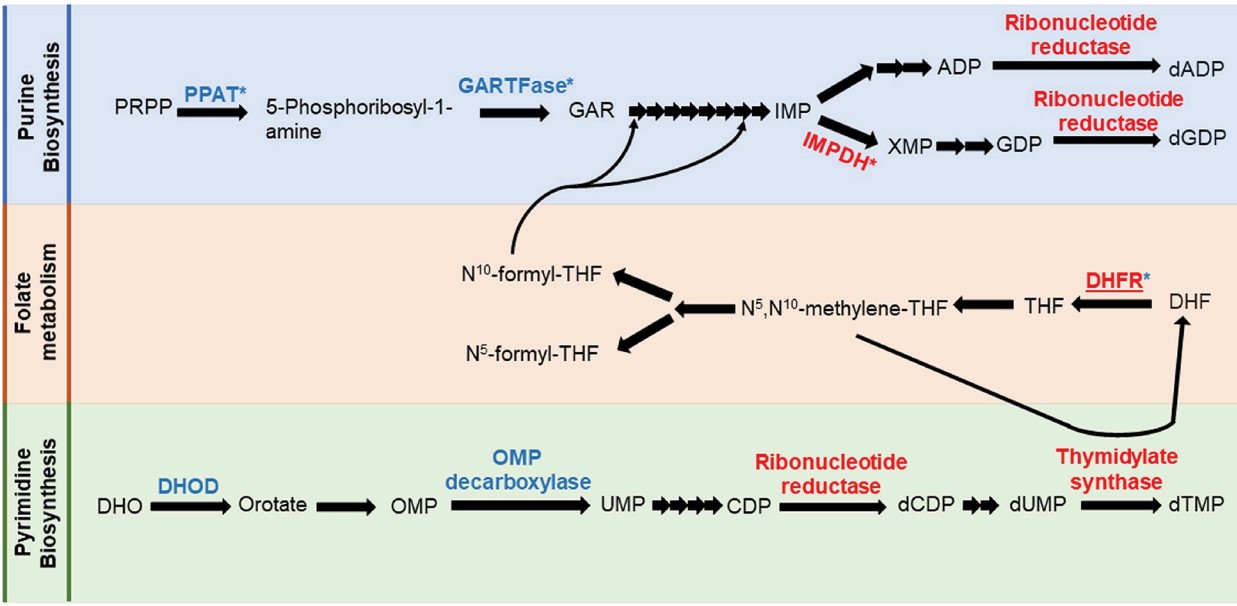

**Fig 7. Inhibition of purine nucleotide biosynthesis impacts differentiation.** Simplified nucleotide biosynthesis pathway of human cells. Enzyme names are colored and only enzymes with putative modulators in our validated hits are shown. Inhibitors of (*) marked enzymes are differentiation inhibitors and others are asexual stage inhibitors. Red-colored enzymes have identified homologs in the *C. parvum* genome while the blue-colored enzymes do not have an identified homolog in the parasite. PRPP: 5-phosphoribosyl pyrophosphate; PPAT: phosphoribosyl pyrophosphate amidotransferase; GARTFase: Glycinamide Ribonucleotide Formyltransferase; GAR: glycinamide ribonucleotide; IMP: Inosine monophosphate; XMP: Xanthosine monophosphate; DHF: Dihydrofolate; DHFR: Dihydrofolate reductase; THF: tetrahydrofolate; DHO: dihydroorotate; DHOD: dihydroorotate dehydrogenase; OMP: orotidine-5'-monophosphate.

phosphorylation was the most upregulated host-cell pathway [27]. These observations collectively form a strong link between *C. parvum* gamont formation and energy production by the host cell. Interestingly, both retested nucleotide biosynthesis inhibitors (mycophenolate and pralatrexate) reduced macrogamont differentiation only when added at 3 hpi and 12 hpi but failed to completely do so when added later (S3 Fig). Perhaps the host ATP pool is sufficiently depleted for inhibition of sexual differentiation only with prolonged exposure to these compounds.

Beyond validating the compounds' effects, our RNA-seq analysis yielded valuable insights about the regulation of *C. parvum* macrogamont differentiation. One is that ribosomal proteins are transcriptionally downregulated and global protein synthesis falls at late stages of *C. parvum* macrogamont differentiation, which was confirmed experimentally by in situ staining of OP-puromycin incorporation in nascent polypeptides (Fig 4A–4D). In *Plasmodium*, ribosomal genes are downregulated during female gametocytogenesis as well as in salivary gland sporozoites compared to midgut oocysts (Fig 4E). In *T. gondii*, ribosomal genes are downregulated in bradyzoites compared to tachyzoites (Fig 4E). Interestingly, in all the three above mentioned differentiation axes, mechanistic studies indicate that post transcriptional repression of gene expression plays a critical role in the differentiation process [18–22]. Coupling these mechanistic studies to the stage-specific transcriptome of these parasites strongly suggests that downregulation of RPs is a transcriptomic signature of translational repression in apicomplexan parasites that also takes place during *C. parvum* macrogamont differentiation. The coregulation of 71 out of 78 RPs may partially be explained by the presence of the DNA motif (A/T/C)GAGAC(A/G) in their promoter regions (Fig 5). RP genes have also been observed to be regulated mostly as a unit in other eukaryotic systems [28–30]. For example, in yeast, there is one or more binding site of the transcription factor Rap1 in almost all of the ribosomal protein gene promoters but that binding site is not unique to ribosomal genes [31, 32]. The system in *C. parvum* seems to be very similar to yeast.

Another significant promoter motif signal that we observed came from a group of proteins that are very likely to be macrogamont stage-specific (Fig 6). Interestingly, this motif is identical to the experimentally validated core binding site of two different Apetala 2 transcription factors [24]. Transcription factors of this class are recognized as key regulators of apicomplexan stage differentiation. In both *Plasmodium* and *Toxoplasma*, multiple Apetala 2 transcription factors often bind to similar DNA motifs and function synergistically or antagonistically. The expression levels of the genes that they regulate are essentially determined by the expression levels of the transcription factors [33–35]. Among the two *C. parvum* Apetala 2 transcription factors that bind to the TGCATGYR motif, cgd2_3490 is significantly downregulated with eight differentiation inhibitor treatments and upregulated with the inducer treatment (Fig 6C). The strongly correlated expression pattern of this transcription factor and its regulome suggests that cgd2_3490 is a key regulator of *C. parvum* macrogamont differentiation. Such insights from the RNA-seq highlight the utility of the differentiation modulators as tools to tease apart the regulation of gamete formation.

In an animal model or in a continuous in vitro culture system, differentiation inhibitors can be used to check if sexual differentiation is an obligatory step following several rounds of asexual replication, i.e., whether the parasite would die vs. continue to grow asexually with a blockade in sexual differentiation. If they die, then inhibiting the sexual stage could be a valid approach for anticryptosporidial drug development. If they continue to replicate asexually, that could lead to the development of a simpler continuous asexual culture system for *Cryptosporidium*. Further in vivo work would help to understand if *Cryptosporidium* sexual differentiation is indeed obligatory, as it appears to be in vitro. However, such studies would require differentiation inhibitors optimized to achieve prolonged chemical exposure in the small

intestinal lumen and tissue, so studies simply using the identified inhibitors *in vivo* would be uninterpretable if negative.

It is important to note that the ReFRAME library is biased towards human protein targets. Applying a similar screening methodology with a less biased library might identify compounds that would more specifically target *C. parvum* proteins. Additionally, there might be other phenotypic patterns that can be caused by compound treatments that we have not captured yet. For example, we have not yet identified a selective gamont maturation inhibitor or a sexual stage-specific inhibitor that would inhibit both macrogamont differentiation and maturation. The utility of such tool compounds would be immense for mechanistic understanding of *C. parvum* development. Even more importantly, our screen relied on the macrogamont marker DMC1 and differences in male and female gametocytogenesis are likely. Additional studies using a microgamont marker to investigate differential impacts of the identified differentiation modulators on male and female gametocytogenesis and future screening to identify male-specific diffentiation modulators would be of great interest.

## Material and Methods

### In vitro *C. parvum* infection model

Human ileocecal adenocarcinoma (HCT-8) cells (ATCC; catalog# CCL-244) were used as host cells for *C. parvum* infection. The cells were passaged in modified RPMI-1640 medium (Gibco, catalog# A10491–01) supplemented with 10% heat inactivated fetal bovine serum (Sigma) and antibiotic solution (120 U/mL penicillin and 120 μg/ml streptomycin; Gibco 15140–122). Cell culture passage number 9 to 36 were used for parasite infection. *C. parvum* oocysts were routinely purchased from Bunchgrass Farm (Deary, ID) and used for up to 6 months. To infect HCT-8 cells, oocysts were excysted in vitro by treating them first with 10 mM HCl (37°C, 10 minutes) and then 2 mM sodium taurocholate (16°C, 10 minutes; Sigma T4009). Host cells are grown to 90% confluency in appropriate plates for infection. Infection media was same as HCT-8 culture media except that it was supplemented with 1% serum instead of 10% and antifungal compound Amphotericin B (0.625 μg/ml; Sigma A2942) was added in addition to the antibiotic solution. Infected cells are incubated at 37°C with 5% $CO_2$.

### Compound screening and dose response validation

Host cells were grown in 384-well plates and infected with 5500 oocysts per well. Following infection, compounds were added at appropriate timepoints using Biotek Precision microplate pipetting system. Cultures were washed with wash solution (0.1% tween 20 in 1× PBS) 3 times before fixing with 4% formaldehyde and permeabilizing with 0.25% triton X-100. 1% BSA was used for blocking before staining with Anti-DMC1 antibody clone 1H10G7 (hybridoma supernatant). Plates were incubatd at 4°C for at-least 24 hours with the primary antibody before washing with the wash solution three times, with five minutes incubation between each wash. Subsequently, the culture was stained with 4ug/ml Alexa Fluor 568 goat anti-mouse IgG (Invitrogen, A-11004) and 1.33 μg/mL FITC-labeled *Vicia villosa* lectin (Vector Laboratories, FL-1231) in 1% BSA for at-least 1 h at 37°C. Finally, 0.09 mM Hoechst 33258 (AnaSpec, catalog# AS-83219) stain was added to stain the nucleus for 10 minutes. Then the wells were washed with wash solution five times, with five minutes incubation between each wash.

The culture plates were imaged with Nikon Eclipse Ti2000 epifluorescence microscope equipped with automated stage movement, using an objective with 20X magnification and 0.45 numerical aperture (NA). 5 by 5 field of views were captured for each well. Images were analyzed with an automated NIH Imagej macro [36] to count the number of nucleus, total parasite and DMC1[+] parasites.

## Quantification of *C. parvum* protein synthesis

HCT-8 cells were grown to 90% confluence in MatTek dishes with fibronectin-coated cover-slips and infected with ~$1x10^6$ oocysts per dish, as above. Cultures were pulsed for 1 h with 50 μM O-propargyl-puromycin (OP-puro)(Tocris, catalog# 7391) prior to fixation at the indicated time points with 4% formaldehyde and permeabilizing with 0.25% triton X-100. Where indicated, the protein synthesis inhibitor cyclohexamide (50 μM) was added 1 h before fixation as a control for staining specificity. Incorporated OP-puro was labeled with azide-AlexaFluor 647 using the Click-IT PLUS kit (Invitrogen, catalog# C10640) according to the manufacturer's instructions. The parasites were then stained with 1.33 μg/mL FITC-labeled *V. villosa* lectin (Vector Laboratories, FL-1231) in 1% BSA for 1 h at 37˚C. Image z-stacks were acquired using a 60X magnification, 1.4 NA oil immersion objective and image spacing of 0.2 μM. Image stacks were deconvolved using Autoquant X software (MediaCybernetics) and 45 iterations.

Relative parasite protein synthesis was quantified from maximum intensity projections using NIH ImageJ and a custom macro that created an image mask from parasite vacuoles labeled with *V. villosa* lectin (green channel) and then output the mean OP-puro (AlexaFluor 647) fluorescence intensity (MFI) of each parasite. Ten image z-stacks were acquired per condition per experiment (n = 3 independent experiments). Parasite numbers were lowest at 12 hpi when the numbers of parasites analyzed per experiment ranged from 549 to 1197. Data are presented as the parasite MFI normalized to MFI at 36 hpi.

## RNA-seq sample preparation and sequencing

HCT-8 cells were grown to 90% confluence on 24-well plates and infected with an appropriate number of excysted oocyst (2, 1, 0.5 and 0.4 million for 18, 36, 48 and 72 hpi samples respectively) in 1 mL of infection media. This difference in MOI ensured higher number of parasite-specific reads in each timepoint without the destruction of the host cell monolayer. Compound or DMSO was added at appropriate time points. Before RNA isolation, wells were washed 3 times with the wash buffer. Total RNA was isolated using the Qiagen RNeasy mini kit (cat #74104). For each biological replicate, total RNA was collected from independent experiments. Messenger RNA enrichment and sequencing libraries were prepared using NEXTFLEX Rapid Directional RNA-seq Library Prep kit bundle (cat # NOVA-5138-10) following the kit manufacturer's protocol. The next-generation sequencing (75 Base pair single end) was performed in the Vermont Integrative Genomics Resource Massively Parallel Sequencing Facility and was supported by the University of Vermont Cancer Center, Lake Champlain Cancer Research Organization, UVM College of Agriculture and Life Sciences, and the UVM Larner College of Medicine. An Illumina HiSeq 1500 was used.

## RNA-seq data analysis

Sequences were demultiplexed and provided in ".fastq" format by the Vermont Integrative Genomic Resource core facility. The sequences were uploaded to the public server of the Galaxy web platform (usegalaxy.org) and all the subsequent analyses were performed in that server [37]. Sequences were quality trimmed using Trimmomatic [38] and then aligned to a combined genome sequence file of Human genome (Gencode release 31, GRCh38, [39]) and *C. parvum* genome (CryptoDB.org release 44, *C. parvum* IowaII, [40]) using HISAT2 [41]. Alignment file was sliced into host part and parasite part using the Slice tool and processed separately in subsequent operations. We used HTseq [42] to determine the number of reads aligned to each gene. Deseq2 was used to analyze differential expression between appropriate

samples [43]. Deseq2 result files were downloaded, relevant values of the genes that met the differential expression cutoff were combined in a Microsoft excel file.

## Pathway analysis and gene expression heatmap generation

Functional annotation enrichment analyses with *C. parvum* genes were performed in DAVID Bioinformatics Resources 6.8 [15, 44]. Appropriate gene ids were copied as gene list and CRYPTODB_ID was selected as Identifier type. All *C. parvum* genes were used as background and all the other options were used as default. For other apicomplexan parasites, "analyze result" was used from an appropriate apicomplexan parasite database after selecting appropriate genes using gene id search [17]. TPM values of *C. parvum* ribosomal proteins were used to generate expression heatmap as well as clustering of samples using shinyheatmap [45]. For clustering, the Euclidean distance metric and the "complete" linkage algorithm were used. To identify similarly expressed genes, the web tool https://software.broadinstitute.org/morpheus/ was used where Pearson Correlation was used to identify genes with similar expression pattern as RP genes.

## *C. parvum* genes' upstream and downstream sequence analysis

1 kb upstream and 1 kb downstream region of all the *C. parvum* genes were uploaded in the usegalaxy.org server. All the *C. parvum* coding sequences were then used as adapter sequences in the cutadapter tool to remove any coding part from the up and downstream sequences of the *C. parvum* genes. For the promoters, we have also trimmed the annotated 5'UTR regions. Such regulatory sequences from the selected genes were used as inputs and regulatory sequences from all the *C. parvum* genes were used as background in the "DREME" tool [46] hosted at "MEME-suite" server [47] to identify significantly enriched motifs. Position of specific motifs at the regulatory sequences of *C. parvum* genes were determined using the "FIMO" tool [48].

## Supporting information

**S1 File. Structure, categorization, and putative mode of action of all the compounds selected as hits from the primary screening.**
(XLSX)

**S2 File. TPM values and log2-fold changes of parasite genes between different treatments and timepoints.** Gene names from the CryptoDB database are included along with the gene IDs. Values are included only if the FDR is below 0.1 in differential gene expression analysis by DEseq2.
(XLSX)

**S3 File. Assigned stage specificity (female-specific, asexual-specific, or unresolved) of *C. parvum* genes based on Tandel dataset [10], and TPM values with log2-fold changes in parasite gene expression with different compound treatments.** Gene names from the CryptoDB database are included along with gene IDs. Values are included only if the FDR is below 0.1 in differential gene expression analysis by DEseq2.
(XLSX)

**S4 File. DAVID functional annotation enrichment analysis of significantly dysregulated genes (fold change ≥ 2 and FDR ≤ 0.1) with differentiation modulator treatment.** The file was created by merging pathway analysis results performed individually for each treatment, upregulated and downregulated gene sets separately. Additionally, the same analysis was

performed for the asexual and female specific genes determined from a previously published dataset [10].
(XLSX)

**S5 File. EupathDB Gene ontology (GO) enrichment analysis of various life cycle stages of different apicomplexan parasites.** In total, comparisons for 9 different axes of differentiation were performed; 5 from *Plasmodium*, 2 from *Eimeria*, and 1 from *Toxoplasma*. For each, both upregulated and downregulated genes are analyzed separately in "cellular component" and "molecular function", culminating in a total of 36 comparisons. The file is a merged file of those 36 results.
(XLSX)

**S6 File. Sexual stage-specific genes used for promoter analysis that were downregulated with differentiation inhibitors but upregulated with differentiation inducer S-1255 treatment.**
(XLSX)

**S1 Fig. Dose response curves of all the screening hits.** 8-point dose response curves of all the primary screening hits in all 4 readouts. IDs correspond to the compound ids from the first column of S1 Table. Duplicate curves (marked with "#") are included for compounds that induce sexual differentiation or development in a dose dependent manner at a concentration range that is not highly inhibitory to asexual replication. For the duplicate curves, relevant sexual stage readouts are excluded for high asexual stage inhibitory concentrations. IDs marked with asterisks (*) are compounds without a determined dose response in one of the readouts due to imaging quality error.
(PDF)

**S2 Fig. Dose response curves of ten differentiation modulators selected for follow-up.** Purchased or resupplied differentiation inhibitors were added at 3 hpi and imaged at 72 hpi (and 48 hpi also for the inducer). Partially overlapping concentration ranges were tested for each of them in 2 to 3 biological replicates. Each point and error bar denotes mean and standard deviation for readings from 4 separate wells for each concentration in each biological replicate. The curves are fitted using the "log(inhibitor) vs. response—Variable slope (four parameters)" model, except for the DMC1 readouts of the macrogamont inducer, where the curve is just a connection between points to highlight the biphasic response. The dotted reference line in the Y axis denotes the concentration selected for RNA-seq experiments.
(TIF)

**S3 Fig. Effect of time of compound addition on differentiation inhibition.** Compounds were added at the indicated time points at the optimal dose to selectively inhibit differentiation, and percent inhibition of macrogamont differentiation (i.e. percent of DMC1$^+$ parasites) was determined at 72 hpi. Data for 2.7 μM nitazoxanide are shown, which was the highest non-toxic concentration tested.
(TIF)

**S4 Fig. Coregulated expression of ribosomal protein genes. A)** Standardized expression of 78 ribosomal protein genes across multiple time points post infection or with compound treatment at 48 hours. Each color represents a gene. Expression levels were normalized by taking the average of expression values for each condition, and then z normalizing it from the expression values across all conditions. **B)** Corelation matrix of 78 ribosomal protein gene standardized expression level. Note the high degree of correlation among most of the genes. **C)** Matrix of P values of the corelations shown in panel B. Most of the correlations are statistically

significant at α = 0.05.
(TIF)

**S1 Table. Number of aligned sequence reads per sample.** Number of RNA-seq reads aligned to the human and *C. parvum* genomes.
(DOCX)

## Author Contributions

**Conceptualization:** Muhammad M. Hasan, José E. Teixeira, Rajiv S. Jumani, Christopher D. Huston.

**Data curation:** Muhammad M. Hasan, Erin E. Stebbins, Christopher D. Huston.

**Formal analysis:** Muhammad M. Hasan, Christopher D. Huston.

**Funding acquisition:** Christopher D. Huston.

**Investigation:** Muhammad M. Hasan, Ethan B. Mattice, José E. Teixeira, Erin E. Stebbins.

**Methodology:** Connor E. Klopfer, Sebastian E. Franco, Christopher D. Huston.

**Project administration:** Christopher D. Huston.

**Resources:** Melissa S. Love, Case W. McNamara, Christopher D. Huston.

**Supervision:** Christopher D. Huston.

**Visualization:** Christopher D. Huston.

**Writing – original draft:** Muhammad M. Hasan, Christopher D. Huston.

**Writing – review & editing:** Rajiv S. Jumani, Case W. McNamara, Christopher D. Huston.

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
