## [Decision Letter · Decision Letter 0]

11 Jan 2024

Dear Dr. Huston,

Thank you very much for submitting your manuscript "CRYPTOSPORIDIUM LIFE CYCLE SMALL MOLECULE PROBING IMPLICATES TRANSLATIONAL REPRESSION AND AN APETALA 2 TRANSCRIPTION FACTOR IN SEXUAL DIFFERENTIATION" for consideration at PLOS Pathogens. As with all papers reviewed by the journal, your manuscript was reviewed by members of the editorial board and by two independent reviewers.

The two reviewers have expressed support for the work and are convinced that it has the potential to reveal new therapeutic compounds inhibiting gamete differentiation. However, they have identified some points that need to be addressed to improve the work.

In light of the reviews (below this email), we would like to invite the resubmission of a revised version that takes into account the reviewers' comments. I trust that you will find the revisions valuable and straightforward.

We cannot make any decision about publication until we have seen the revised manuscript and your response to the reviewers' comments. Your revised manuscript is also likely to be sent to reviewers for further evaluation.

Sincerely,

Dominique Soldati-Favre

Section Editor

PLOS Pathogens

Dominique Soldati-Favre

Section Editor

PLOS Pathogens

Kasturi Haldar

Editor-in-Chief

PLOS Pathogens

orcid.org/0000-0001-5065-158X

Michael Malim

Editor-in-Chief

PLOS Pathogens

orcid.org/0000-0002-7699-2064

Reviewer's Responses to Questions

**Part I - Summary**

Reviewer #1: Hasan and colleagues describe the screen of a 12,000-compound library to identify compounds which interfere with C. parvum gamete differentiation in culture. This approach is novel and extends the power of high-throughput screening to the rarely investigated sexual phase of the Cryptosporidium life cycle. Since sexual differentiation is an obligatory phase of the life cycle, Hasan et al. logically argue that inhibitors of gamete differentiation could serve as chemical probes and lead to the discovery of new therapeutic compounds. The manuscript is generally well written and informative.

Reviewer #2: The authors have generated a novel high-throughput small molecule screen that permits them to identify compounds that specifically target the sexual differentiation/gamete formation portion of the Cryptosporidium parvum lifecycle. Cryptosporidium is a significant global human and veterinary diarrheal pathogen that is most prevalent in the very young and immunosuppressed. Currently there is only one approved therapeutic, a drug that targets all life cycle stages, but unfortunately is not approved for use in the most affected populations. The appeal of therapeutic compounds that target this developmental stage is the fact that there is still a bit of controversy as to whether or not Cryptosporidium is obligately sexual in each ~72 hour portion of its life cycle both in vivo and in vitro. Recent work from the Striepen lab revealed a lifecyle for Cryptosporidium that differs from the previously described lifecycle in which only a portion of the parasites enter the sexual cycle. If, as is now appears to be the case, all parasites terminally differentiate into sexual forms in order for the parasite to replicate and complete the lifecycle, a compound that could target this stage could halt infection. Such a compound, as the authors point out, would also be very useful in the laboratory as well and could definitively prove if there is any asexual replication beyond the demonstrated 3 cycles prior to gamete formation. Additionally, as a result of the experiments performed in this work, there is now a list of 65 co-regulated genes that are specifically associated with sexual differentiation, primarily macrogamont formation. Not only are these useful markers, but analyses of their function provided insights into the biology of sexual differentiation in Cryptosporidium and also, possibly gene regulation. This study has a solid experimental design and the experiments are rigorous. I take exception to a few of the interpretations of the results but overall this work represents a significant step forward on several fronts (screens for compounds that affect a different life cycle stage than the one normally targeted in drug screen and the insights gained from the follow-up on the patterns of altered gene RNA expression in parasites affected by particular compounds). The manuscript could be clearer throughout that it is macrogamont differentiation that is being studied and not "sexual differentiation" writ large.

**Part II – Major Issues: Key Experiments Required for Acceptance**

Reviewer #1: On line 222 it is stated that the experiment was duplicated, but the Methods section does not describe how duplication was performed. It does make a difference if 2 wells from a same plate were exposed to a compound or the entire experiment was performed with truly independent replicates. The “biological replicate” implies the latter.

The statement that the expression of ribosomal proteins is correlated is not apparent from Fig. 4E. A correlation matrix might be a better way to illustrate this point and a statistical test is needed.

Reviewer #2: I have two concerns:

1) There is the assumption in the manuscript that if the levels of mRNAs for ribosomal proteins decline that this is akin to a state of reduced translation. This may very well be the case, but reduced translation is not explicitly examined or proven. We do not know how long the messages and the proteins they encode are stable. I suggest that the author either prove this point, or, tone down the interpretation to the proven reduced levels of transcription for particular categories of genes. Either way, the finding is significant and as the authors point out, the finding is similar to observations with other apicomplexans. This is surprising given the evolutionary distance between the members of this phylum.

2) The upstream motif analysis for putative regulatory elements in the coordinately regulated ribosomal protein genes generated a result that is contradictory to a previous result, Oberstaller et al., BMC Genomics 2013 14:516. In the prior analysis, C. parvum ribosomal protein putative promoter regions were found to be enriched for E2F-like and GAGA motifs, contrary to the motifs found in P. falciparum. The major difference between the two analyses is the the definition of upstream region. In the 2013 analysis only upstream regions beginning with the CDS "ATG" that did not encroach into a neighboring gene on either strand were used, and in the analysis reported here, 1 kb of upstream sequence was used. 1 kb almost always overlaps with another gene given the density of this tiny genome. A newly annotated C. parvum genome sequence that includes annotated UTRs was just released by my group and it will permit the best possible analysis since the transcription start site for most transcripts is now known and the UTRs are annotated. See: Genbank nucleotide assembly GCA_035232765.1 and bioRxiv 10.1101/2023.06.13.544219v1.full.pdf for details. It would be ideal to perform this analysis again with the more defined upstream region and also to examine the expression profiles for the several E2F and DP1 transcription factors. The results might change your title.

**Part III – Minor Issues: Editorial and Data Presentation Modifications**

Reviewer #1: Given the essential role the immunofluorescent detection of DMC1 plays in the assay, I was wondering about the absence of an uninfected control monolayer in Fig. 1A and 1B. Further regarding Fig. 1A, if the lectin stains any life cycle stage and anti DMC1 antibody reacts with macrogamonts, why do the two signal not overlap in the 72 h merged image? If the original micrographs indicate otherwise, perhaps this should be stated in the text or the legend.

Many blue dots in Fig. 1D are within the 2 SD boundary and yet, according to the legend, were selected for follow-up screens. Why? Also according to the legend, the y axis in 1D is in SD deviation, but the axis label in the figure says % inhibition. Please clarify.

Fig. 3E bar graph, please indicate what constitutes 100%; percentage of what?

Line 528; Please indicate why for the RNA-Seq analysis cell monolayer were infected with different oocyst doses and whether this experimental design introduces an uncontrolled variable, namely oocyst:host cell ratio.

Line 310 onward describing the relevance of energy metabolism should probably move to the Discussion. Redundant text in the Discussion, such as lines 388-401 should be removed.

Line 128, “..but were not visible..” (verb missing)

Line 136; Do you mean “sexual” differentiation?

Line 157; “.. that detectably modulated..”

Lines 285, 309 and Fig. 4E legend; Do you mean “differentially regulated” as opposed to dysregulated? Dysregulation implies an anomaly.

Reviewer #2: 1. mRNA-seq and RNA-seq both appear throughout the manuscript and this was a bit confusing. I had the sense, perhaps incorrectly that mRNA-seq was being used because a poly-A step is included in the library prep and the focus was on protein encoding genes. However Cryptosporidium has a lot of ncRNAs and most have poly-A tails. If it is the author's intent to make a distinction, please explicitly state how each term will be used.

2. Please state the quantity of reads generated and the percentage of Cryptosporidium transcripts for each library. This can be a supplementary table. The TPM numbers for Cryptosporidium in Supplementary file 2 are decent, but they not high. This is not a surprise. Did you have a cutoff for expression levels to be able to perform the expression analyses? it is not clear from the methods. It appears that NovaSeq was used, is this correct?

3. The methods state that 1 kb upstream was analyzed for motifs but in Supplementary file 6 many shorter promoter lengths are listed. Which sequence was used? 1 kb or the lengths in file 6?

4. The manuscript does not have a data accessibility section. Have the RNA-seq reads that were generated been deposited in the SRA? Please provide the bioproject and accession numbers. I realize the data for other species come from VEuPathDB.org, but they are not an archive for new data5. A potential item to add to the discussion is the observation in Figure 3C that the number of upregulated genes is surprisingly uniform but the number of down-regulated genes is highly variable. Do the authors have any thoughts on this?

5. Figure 4A. Please clarify if these data are only for C. parvum

6. Figure 7 legend, last line, "transcript factors that bind a" without the s on binds

7. Lines 44-45 translational repression is not experimentally validated

8. Line 81 " Cryptosporidium sexual differentiation"

9. Line 116 mechanism? perhaps pathway or regulation?

10. Table 1 - Just curious. Given the recent results from the IMPDH knockout in Cryptosporidium why do you think mycophenolate still came through the screen as strongly as it did? perhaps an off target host effect?

11. Line 268 all sexually differentiated, or just macrogamonts? I thought the screen could only detect the female gamete.

12. Line 287 VEuPathDB.org

13. Line 327 perhaps move this to the discussion?

14. Lines 353-354, not necessary. Gene regulation is often regulated combinatorially either by differing numbers of the same TF site or combinations of TF sites or both. please consider rephrasing.

15. Lines 416-417 and 425, this is not proven, please revise.

16. Lines 455-457 I think a reference is missing, these results are not from this study.

17. Line 546 CryptoDB.org

18. Line 547 add italics

19. Lines 566-574, please define how UTR and promoter regions were determined/are defined

20. Should the title of the manuscript make it clearer that the topic is macrogamont differentiation?

PLOS authors have the option to publish the peer review history of their article (what does this mean?). If published, this will include your full peer review and any attached files.

Reviewer #1: No

Reviewer #2: **Yes: **Jessica Kissinger

Figure Files:

Data Requirements:

Please note that, as a condition of publication, PLOS' data policy requires that you make available all data used to draw the conclusions outlined in your manuscript. Data must be deposited in an appropriate repository, included within the body of the manuscript, or uploaded as supporting information

---

## [Decision Letter · Decision Letter 1]

8 Apr 2024

Dear Dr. Huston,

We are pleased to inform you that your manuscript 'CRYPTOSPORIDIUM LIFE CYCLE SMALL MOLECULE PROBING IMPLICATES TRANSLATIONAL REPRESSION AND AN APETALA 2 TRANSCRIPTION FACTOR IN MACROGAMONT DIFFERENTIATION' has been provisionally accepted for publication in PLOS Pathogens.

Before your manuscript can be formally accepted you will need to address the minor points raised by Rev#2 and to complete some formatting changes, which you will receive in a follow up email. A member of our team will be in touch with a set of requests.

Best regards,

Dominique Soldati-Favre

Section Editor

PLOS Pathogens

Dominique Soldati-Favre

Section Editor

PLOS Pathogens

Michael Malim

Editor-in-Chief

PLOS Pathogens

orcid.org/0000-0002-7699-2064

Reviewer Comments (if any, and for reference):

Reviewer's Responses to Questions

**Part I - Summary**

Reviewer #2: The clarity and precision of the revised manuscript are greatly improved.

**Part II – Major Issues: Key Experiments Required for Acceptance**

Reviewer #2: My concerns have been well addressed. I appreciate the author's work to more precisely define the promoter regions.

**Part III – Minor Issues: Editorial and Data Presentation Modifications**

Reviewer #2: This is very minor: Titles in Supplemental Table 1 could be improved, i.e. "aligned reads" as opposed to "aligned read" in two columns, italics for C. parvum and perhaps add the word genome in "percent aligned to the parasite/with human"

Line 515, add a space between 10 and mM HCL

Line 516, add a space between 2 and mM sodium and degree symbol is missing in many places in this paragraph

Line 531, add a space between 4 and ug/ml

Reference 24 and 33 appear to be duplicates

Italics missing from Figure 3 legend in several places

Italics missing from Figure 4 Title.

PLOS authors have the option to publish the peer review history of their article (what does this mean?). If published, this will include your full peer review and any attached files.

Reviewer #2: No

---

## [Editor Report · Acceptance letter]

19 Apr 2024

Dear Dr. Huston,

We are delighted to inform you that your manuscript, " CRYPTOSPORIDIUM LIFE CYCLE SMALL MOLECULE PROBING IMPLICATES TRANSLATIONAL REPRESSION AND AN APETALA 2 TRANSCRIPTION FACTOR IN MACROGAMONT DIFFERENTIATION," has been formally accepted for publication in PLOS Pathogens.

Best regards,

Michael Malim

Editor-in-Chief

PLOS Pathogens

orcid.org/0000-0002-7699-2064